# Gender Differences and Immunotherapy Outcome in Advanced Lung Cancer

**DOI:** 10.3390/ijms222111942

**Published:** 2021-11-04

**Authors:** Tiziana Vavalà, Annamaria Catino, Pamela Pizzutilo, Vito Longo, Domenico Galetta

**Affiliations:** 1ASL CN1, Struttura Complessa di Oncologia, Ospedale di Savigliano, Via Ospedali 9, 12038 Savigliano, Italy; 2Thoracic Oncology, IRCCS Istituto Tumori “Giovanni Paolo II”, Viale O. Flacco 65, 70124 Bari, Italy; a.catino@oncologico.bari.it (A.C.); pamela.pizzutilo@gmail.com (P.P.); v.longo@oncologico.bari.it (V.L.); galetta@oncologico.bari.it (D.G.)

**Keywords:** gender differences, lung cancer, immunotherapy

## Abstract

In developed countries, lung cancer is the leading cause of cancer-related death in both sexes. Although cigarette smoking represents the principal risk factor for lung cancer in females, the higher proportion of this neoplasm among non-smoking women as compared with non-smoking men implies distinctive biological aspects between the two sexes. Gender differences depend not only on genetic, environmental, and hormonal factors but also on the immune system, and all these aspects are closely interconnected. In the last few years, it has been confirmed that the immune system plays a fundamental role in cancer evolution and response to oncological treatments, specifically immunotherapy, with documented distinctions between men and women. Consequently, in order to correctly assess cancer responses and disease control, considering only age and reproductive status, the results of studies conducted in female patients would probably not categorically apply to male patients and vice versa. The aim of this article is to review recent data about gender disparities in both healthy subjects’ immune system and lung cancer patients; furthermore, studies concerning gender differences in response to lung cancer immunotherapy are examined.

## 1. Introduction

A sex-based dimorphism is increasingly emerging in tumor pathology. Men present an approximately two-times higher risk of mortality from all cancers than women; different outcomes based on sex are particularly relevant for lung, melanoma, larynx, esophagus, and bladder cancers [1]. Gender differences in this context depend not only on biological, environmental, and hormonal factors but also on immune system, and all of these aspects are strictly interconnected [2,3,4]. 

In the last few years, it has been confirmed that immune system plays a fundamental role in cancer evolution and response to oncological treatments, specifically immunotherapy, with documented distinctions between men and women [5].

Consequently, in order to correctly assess cancer responses and disease control considering only age and reproductive status the results of studies conducted in female patients would probably not categorically apply to male patients and vice versa.

At present, in most of pre-clinical and clinical studies, female sex is still underrepresented as compared with males, but integration of data from both sexes is fundamental to understand any gender impact on disease evolution as well as for driving the path to distinct, sex-based diagnostic and therapeutic strategies particularly when immune approaches are under evaluation [6,7].

The aim of this article is to review recent data about gender disparities in both healthy subjects’ immune system and lung cancer patients; furthermore, studies concerning gender differences in response to lung cancer immunotherapy are examined.

## 2. Sex Differences in Immune System, Hormonal Influences, and Impact of Smoking Status in Healthy Subjects

Women commonly produce stronger innate and adaptive immune responses than men, with a lower prevalence of infections or cancer but a higher incidence of systemic autoimmune diseases [2].

Genes with significant roles in the regulation of immune response, such as those that encode for *IL-2 receptor gamma subunit*, *toll-like receptor (TLR)-7*, *TLR-8*, *CD40L*, and the *fork-head box P3 (FOXP3**)*, are located on the X chromosome. The production of cytokines and chemokines by innate immune cells differs between the two sexes as well as cellular activities associated with innate immunity. Notably, neutrophils from males subjects produce a greater amount of tumor-necrosis factor (TNF) than females; furthermore, men show higher natural killer (NK) cell rates and have a more elevated number of innate lymphoid cells (ILCs—innate-like lymphocytes able to regulate immune responses through effectors cytokines) than women. On the contrary, dendritic cells and macrophage activities are enhanced in females, and antigen-presenting cells (APCs) from females are more efficient in presenting peptides than APCs in males [2,8]. 

Additionally, in the context of adaptive immunity, numerous key immune-related genes are located on the X chromosome. Sex differences in lymphocyte subsets are described in Asian, European, and African populations; women show a greater antibody response than men, with higher basal immunoglobulin levels and B-cell numbers. This last evidence could be due to a significant up-regulation in B cells in females as compared with males, as described in a global analysis of B-cell gene-expression signatures performed by Fan and Coll on a small group of both healthy subjects and patients with systemic lupus erythematosus disease [9,10,11,12]. The activity and distribution of CD4+ T-cell subsets also differ between the two sexes. In fact, females show higher CD4+ T-cell counts and produce higher levels of IFN-γ than males, and women present also higher CD4/CD8 ratios than age-matched males with an higher number of activated CD4+ T, CD8+ T cells and proliferating T cells in peripheral blood; on the contrary, in males, higher CD8+ T-cell ratios are reported as well as higher numbers of T regulatory cells (Tregs) in healthy adult males compared to women (Table 1) [2,8,10,11].

Many aspects of functional activity of innate immune cells and downstream adaptive immune responses are influenced by hormonal mediators, such as differentiation, maturation, and functions of dendritic cells, neutrophils, NK cells, macrophages, and B and T lymphocytes. Particularly, Estrogen Receptor (ER)alfa and ERbeta, which are also involved in lung tumorigenesis in both sexes, basically display a differential expression among immune cells subsets: ERalfa is highly expressed in T cells, and ERbeta is up-regulated in B cells; this has been hypothesized on the basis of experimental observations suggesting that treatments of either humans or mice with estrogen (such as 17-beta-estradiol) increase neutrophil count in blood and lungs, respectively [2,12]. Non-classical ER signaling also occurs in immune cells, enabling interactions between ERs with estrogen response elements (EREs) independent transcription factors, including nuclear factor-kappa B (NFκB), specific protein 1 (SP-1), and activator protein 1 (AP-1); conversely, androgens may repress the activity of NFκB to control inflammation [13]. EREs and androgen response elements (AREs) have been documented in several innate immunity genes promoters; remarkably, in female T cells, half of activated genes comprises EREs in their promoters, emphasizing the findings that sex steroids may directly influence immune responses, particularly leading to stronger inflammatory and cytotoxic T-cell responses in females [14].

Both innate and adaptive immunity are susceptible to cigarette smoke; recent studies have shown that cigarette smoke starts MAPK signaling pathways, which sequentially regulate transcription factors (TFs) activation and modify DNA-binding capacity of more than 20 TFs, including NFκB. Functional alterations of TFs contribute to transcriptional changes of their target genes, including inflammatory cytokines and chemokines (Table 2) [15,16,17]. However, even if scientific studies describe various cellular mechanisms responsible for immune regulation, the distinct impact of cigarette smoke and different reaction of immune system according by gender is largely unknown as well as the exact molecular pathways underlying smoking-associated immune pathology.

For now, it has been suggested that women show greater susceptibility to cigarette smoke-induced DNA damage as well as higher levels of DNA adducts than males [18]. On the basis of this hypothesis, Pinto and Coll evaluated the expression and mutational status in DNA repair-involved genes without observing any difference between sexes; in a subsequent global evaluation, they described some gene sets, particularly immune ones, differentially enriched in women [19]. Further studies are needed to better clarify the impact of cigarette smoke on immune reactions and consequently the impact on lung cancer risk in both sexes.

## 3. Sex Differences in Immune System of Cancer Patients

Sex variability in the immune system could justify gender disparities in cancer incidence, mortality, and treatments responses [20]. In particular, sex disparities in lung cancer oncogenesis have been evidenced as linked to epithelial STAT3 deletion in mice with mutant Kras: in males, the lack of epithelial STAT3 induced lung tumorigenesis via enhanced IL-6 signaling and neutrophilic inflammation, which was inhibited in females by estrogen signaling [21]. However, all these findings are not conclusive since estrogens present a bi-potential effect and with low doses are able to enhance production of pro-inflammatory cytokines (such as IL-6 but also IL-1 and TNF), while at high concentrations, this production is reduced [2,22].

Certainly, these studies confirm that estrogens modulate inflammatory cytokine secretion by macrophages and neutrophils, and this could potentially reduce cancer risk in females [23,24]. In addition, even if a higher incidence of non-small cell lung cancer (NSCLC) is currently reported in women due to intensification of smoking habits in the last few years, a better prognosis of NSCLC in the female sex could be due to other (not better known at present) immune distinctions, such as different immune gene sets enrichments in females with NSCLC when compared to males [23]. As previously described, hormones may impact on anti-tumor immunity and on treatment responses. B7-homolog 1 (B7-H1), also known as Programmed death-ligand 1 (PD-L1), a co-signaling molecule richly expressed on APCs, contributes to tumor immune evasion and induces Treg function, but it can be modulated in an estrogen-dependent manner [24]. PD-L1 expression could also be controlled by several X-linked microRNAs, suggesting that its role as a confident predictive biomarker, from a gender point of view, remains controversial [25].

Both in men and women, tumor cells, to invade and spread, need to escape immune surveillance by loss of MHC molecules or up-regulation of immune checkpoint elements, which usually modulate amplitude and duration of T-cell responses on cell surfaces; indeed, antibodies for immune checkpoint blockade (ICB) are administered to stimulate inactive and/or exhausted T cells to react against cancer [26]. However, according to current data, only 10–40% of patients in both sexes benefit from ICB, still for unclear reasons. One of possible explanations could be the observation that tumor cells in females face more efficient immune surveillance mechanisms and are exposed to powerful immune-editing processes to become metastatic. These conditions induce an increased ability of cancer cells to evade immune surveillance evolving into less immunogenic, advanced cancers, which may finally become resistant to immunotherapy approaches [27].

In this context, PD-L1 and Cytotoxic T-lymphocyte protein 4 (CTLA-4) pathways, which are significantly involved in tumor-induced immune control, are now druggable and still under evaluation to elucidate gender differences also in lung cancer. However, further studies about these fundamental issues are needed.

## 4. Current Studies about Immunotherapy Approaches in Lung Cancer According to Sex

ICB approaches have shown to improve survival across multiple cancer types; however, there is still a huge debate principally based on multiple meta-analyses ranging among different tumor types regarding a possible different outcome in male and female cancer patients undergoing immunotherapy.

Botticelli and Coll selected 36 phase II–III Clinical Trials published up to June 2017 in which anti-CTLA-4, anti-PD-1, and anti-PD-L1 were studied. Nine of them were finally considered, and five were phase II–III or III enrolling NSCLC patients. No significant benefit with anti-PD-1 in OS nor in progression-free-survival (PFS) in males vs. females (HR = 0.72, 95% CI, 0.64–0.83 vs. HR = 0.81, 95% CI, 0.70–0.94, *p* = 0.285 and HR = 0.66, 95% CI, 0.52–0.82 vs. HR = 0.85, 95% CI, 0.66–1.09, *p* = 0.158, respectively) was evidenced. Notably, the authors did not include in their final meta-analysis anti-PD-L1 treatments because of lacking data, while the only two trials with anti-CTLA4 therapies are not commented in this article as they had only enrolled patients with melanoma.. This study has important limitations, such as trials’ heterogeneity and different cancer types considered, absence of records about hormonal and PD-L1 status according to sex, and different cut-offs of biomarkers expression [28]. Therefore, results from this study can only suggest further investigations about this topic.

In a meta-analysis conducted by Pinto and Coll, only NSCLC patients were included: five phase III studies comparing anti-PD1 inhibitors versus chemotherapy, two studies with pembrolizumab versus chemotherapy (KEYNOTE 010 and KEYNOTE 024), and three with nivolumab versus chemotherapy (CHECKMATE 017, CHECKMATE 026, CHECKMATE 057). A total of 1028 female and 1435 male patients were evaluated. In male patients, an overall HR = 0.76 (95% CI, 0.68–0.86, *p* < 0.00001) favoring anti-PD1 inhibitors was observed; however, a significant heterogeneity between studies (95% CI, 0.68–0.86, *p* = 0.0001) was also evidenced. For female patients, there was no clear benefit from nivolumab or pembrolizumab when compared with chemotherapy (HR = 1.03; 95% CI, 0.89–1.03, *p* = 0.69), without significant heterogeneity between the cohorts (*p* = 0.45; I^2^ = 33%). This meta-analysis showed a 24% reduction in the risk of disease progression in men treated with anti-PD1 inhibitors (nivolumab and pembrolizumab), while women presented a smaller benefit. However, also in this study, the interpretation was limited, particularly for males-related information, by a relevant trials’ heterogeneity as well as a difference in cut-offs of biomarkers expression: KEYNOTE 010, KEYNOTE 024, and CHECKMATE 026 trials included PD-L1 tumor-expression positivity as an inclusion criteria, while CHECKMATE 017 and CHECKMATE 057 trials included patients with NSCLC regardless of their PD-L1 status [29].

In the meta-analysis published by Wu and Coll, 11 phase II, II/III, and phase III trials were examined to assess CTLA-4 or PD-1 inhibitors efficacy versus chemotherapies or other therapies. A total of 6096 patients were considered, including 2192 patients with NSCLC (four trials). A better PFS (HR = 0.57; 95% CI, 0.43–0.71; *p* < 0.001 HR = 0.71; 95% CI, 0.52–0.91; *p* < 0.001) was observed in males versus females treated with immune checkpoints inhibitors (ICIs) as well as an improvement in OS (HR = 0.62; 95% CI, 0.53–0.71, *p* < 0.001 vs. HR = 0.74; 95% CI, 0.65–0.84; *p* < 0.001), respectively. However, this difference was not significant in NSCLC cohort for PFS (*p* = 0.07) and OS (*p* = 0.373). Of note, as for the previously cited meta-analyses, this one by Wu was also biased by trials’ heterogeneity, different cancer types considered, and lacking data about hormonal and PD-L1 status according to sex [27].

Grassadonia et al. evaluated, in their meta-analysis, 12,635 patients with advanced cancer in 21 randomized control trials (8410 males and 4225 females) comparing CTLA-4 or PD-1 inhibitors versus chemotherapies or other therapies. Of them, 10 trials considered NSCLC patients, and 1 included small-cell lung carcinomas. PFS calculated on eight trials enrolling NSCLC patients was longer in men than in women (HR = 0.67, 95% CI, 0.55–0.80, *p* < 0.001 and HR = 0.77, 95% CI, 0.57–1.05, *p* = 0.100, respectively). Subgroup analyses by specific ICIs showed similar OS in males and females for both anti-PD-1 and PDL-1. Anti-CTLA-4 treatment was associated with longer OS in men only (HR = 0.77, *p* < 0.012) except for melanoma [30].

Similarly, Conforti et al. considered, in their meta-analysis, 11,351 patients (67% men and 33% women) with advanced cancers. Of the overall cohort, 3482 (31%) were NSCLC patients, while 1478 of them were women. The pooled OS HR was 0.72 (95% CI, 0.65–0.79) in men and 0.86 (95% CI, 0.79–0.93) in women treated with ICIs versus, respectively, men and women in the control groups. The difference in efficacy between the two sexes treated with ICIs was significant (*p* = 0.0019) [31,32]. In both of the aforementioned meta-analyses, despite the large number of patients analyzed, a lower number of women was finally considered (in half of the examined trials, women represented no more than half of the entire population). This is a potential limitation to observe a significant interaction between sexes and ICIs efficacy as well as trials’ heterogeneity, different cancer types considered, and lacking data about hormonal and PD-L1 status according to sex, as already described for the previous meta-analyses.

Wallis and Coll evaluated, in their meta-analysis, a total of 23 studies. Patients included were 13,721 (67.9% men) with advanced cancer. Unlike the Conforti study, this meta-analysis did not show any difference in terms of OS following immunotherapy between the two sexes, with a benefit for both men (HR = 0.75, 95% CI, 0.69–0.81, *p* < 0.001) and women (HR = 0.77, 95% CI, 0.67–0.88, *p* = 0.002). These contradictory results may be due to a different study selection in terms of type of ICIs and regimens (for instance, Wallis et al. also included, unlike Conforti, atezolizumab in their final evaluation) and to an update with seven additional trials [33].

Wang et al. evaluated 9583 advanced lung cancer patients from 15 randomized controlled trials (68.5% men and 31.5% women). The authors reported a significant PFS benefit, based on 10 of 15 trials in both men (HR = 0.67, 95% CI, 0.58–0.77) and women (HR = 0.73, 95% CI, 0.56–0.95) who received ICIs vs. standard therapies. Particularly, a longer PFS for anti-PD-1 treatments (HR = 0.71, 95% CI, 0.58–0.88), for anti-PD-L1 ones (HR = 0.64, 95% CI, 0.56–0.74), monotherapy (HR = 0.72, 95% CI, 0.56–0.92), and combination (HR = 0.64, 95% CI, 0.57–0.71) was evidenced in male NSCLC patients. In females, PFS benefit was described for anti-PD-L1 therapies (HR = 0.56, 95% CI, 0.45–0.69) and combination ones (HR = 0.53, 95% CI, 0.43–0.64) but not for anti-PD-1 treatments (HR = 0.83, 95% CI, 0.57–1.20) or monotherapy (HR = 1.02, 95% CI, 0.84–1.23). Pooled results showed a reduced risk of death for both male (HR = 0.76, 95% CI, 0.71–0.82, *p* < 0.001) and female patients (HR = 0.73, 95% CI, 0.58–0.91, *p* = 0.007) following administration of ICIs. Concerning OS, there was a benefit in male patients for anti-PD-1 therapies (HR = 0.73, 95% CI, 0.67–0.80) and anti-PD-L1 ones (HR= 0.80, 95% CI, 0.69–0.92), while in females, OS benefit was evidenced for anti-PD-1 treatments (HR = 0.69, 95% CI, 0.52–0.93) but not for anti-PD-L1 ones (HR = 0.69, 95% CI, 0.44–1.07). Monotherapy with ICIs was suggested to induce an OS benefit compared with combination therapy (combination of ICIs or ICIs plus chemotherapy) for both sexes. No survival benefit was evidenced for CTLA-4 inhibitors treatments in both men and women. In this study, as for previous ones, multiple limitations are related to subgroup HRs of OS and PFS rather than individual data, studies heterogeneity in female patients (I^2^ = 76.1%, *p* < 0.001), lack of sex-subgroup data, and furthermore, for most of the included randomized controlled trials, OS and PFS data were not mature to provide consistent results [34].

Finally, Dafni et al., in their network meta-analysis including 9236 metastatic NSCLC patients, compared the efficacy of treatments with at least one ICI with or without chemotherapy as a first-line approach. Among the examined variables, the authors also considered sex and evidenced that the same treatment combinations showed a benefit in PFS and OS in both males and females, while pembrolizumab plus chemotherapy appeared more active in women than men [35].

At the present time, considering that all previous observations emerged from meta-analyses underpowered to explore the effect of gender disparities on outcomes, there are not conclusive data about gender differences in response to immune therapies.

For now, isolated, sex-based subgroup analyses are simply hypothesis generating [36].

It can be argued that ICIs may result more effectively in men than women probably for the higher antigenicity of their cancer cells, while in females, more efficient escape mechanisms of cancer cells due to a stronger immune system could induce greater resistances to ICIs. For that reason, improving the immune environment in male patients and antigenicity of tumors in female ones could be a useful strategy deserving of testing in future prospective studies.

## 5. Conclusions

Despite a growing amount of literature data illustrating sex-based differences in immune system and responses to oncological and immune treatments for cancer patients, an insufficient number of articles analyze data by sex in a prospective and pre-planned manner. In the clinical research setting, in order to specifically clarify the effect of gender on cancer treatment outcomes, it is now primarily useful to weight potential confounding factors, such as race, tumor stage, histological type, and molecular biomarkers, together with smoking habits and menopausal status. This, particularly in the innovative immune-oncology branch and thoracic oncology, should be in the very near future a fundamental prerequisite for improving the knowledge of lung cancer evolution mechanisms and consequently developing more personalized approaches in both sexes.

## Figures and Tables

**Table 1 ijms-22-11942-t001:** Main gender differences in innate and adaptive immunity [2,8,10,11].

Enhanced in Females	
Innate immunity	Adaptive immunity
Neutrophils phagocitic capacity	CD4+ T-cell count
Macrophagic activation	CD4/CD8 T-cell ratio
Macrophagic phagocitic capacity	T-cell proliferation
APC efficiency	Activated T-cell count
Dendritic cells activities	T-cell cytotoxicity
Toll-like receptors gene expression pathway	B-cell count
	Antibody production

**Table 2 ijms-22-11942-t002:** Influence of smoking status on human immune system.

Innate Immunity	Adaptive Immunity
Increased neutrophils count	Increased T-cell count
Reduced neutrophils activity Reduced APC efficiency	Reduced global T-cell activity Increased CD4+ T-cell activity
	Increased auto-reactive B-cell activity
	Reduced circulating immunoglobulins

## Data Availability

Not applicable.

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
