# Peer review of "Gender Differences and Immunotherapy Outcome in Advanced Lung Cancer"

_ijms, 2021, doi:10.3390/ijms222111942_

Round 1

Reviewer 1 Report

The topic is important and interesting. Although many articles are devoted to it, not all questions have been answered and not all aspects have been discussed yet. The manuscript provides a fairly good comprehensive overview.

Some concerns and suggestions:

  1. As the most important target of the article is the immune response, I propose to extend Table 1 with other markers and to indicate the literature sources for each of them inside of Table.
  2. The aim of the article is the gender difference in lung cancer. Therefore, Table 2 should be redesigned to show gender differences and augmented.
  3. Some publications containing information and references related to this article are suggested for the authors' review:

Frega S, Dal Maso A, Ferro A, Bonanno L, Conte P, Pasello G. Heterogeneous tumor features and treatment outcome between males and females with lung cancer (LC): Do gender and sex matter? Crit Rev Oncol Hematol 2019; 138: 87-103.

MacRosty CR, Rivera MP. Lung cancer in women: a modern epidemic. Clin Chest Med 2020; 41(1): 53-65.

Mederos N, Friedlaender A, Peters S, Addeo A. Gender-specific aspects of epidemiology, molecular genetics and outcome: lung cancer. ESMO Open 2020; 5(Suppl 4): e000796.

Author Response

Thank you for the opportunity to review this article. I enjoyed very much during reading this munscript. Well organised with comprehensive review for all the available and important data in this manner.

A small comment for the authers, would be nice if you add in the conclusions, based on your review what kind of prospective trials should be done in the future to find out the effect of gender on cancer treatment outcomes and due you suggest to test any specific markers in this manner 

We thank the Reviewer for this suggestion. We have added a comment in the Conclusion paragraph as suggested

Reviewer 2 Report

Thank you for the opportunity to review this article. I enjoyed very much during reading this munscript. Well organised with comprehensive review for all the available and important data in this manner.

A small comment for the authers, would be nice if you add in the conclusions, based on your review what kind of prospective trials should be done in the future to find out the effect of gender on cancer treatment outcomes and due you suggest to test any specific markers in this manner 

Author Response

The topic is important and interesting. Although many articles are devoted to it, not all questions have been answered and not all aspects have been discussed yet. The manuscript provides a fairly good comprehensive overview.

Some concerns and suggestions:

  1. As the most important target of the article is the immune response, I propose to extend Table 1 with other markers and to indicate the literature sources for each of them inside of Table.     We thank the Reviewer for this suggestion. We have modified Table 1 by adding more immunity markers and indicating literature source, as suggested.
  2. The aim of the article is the gender difference in lung cancer. Therefore, Table 2 should be redesigned to show gender differences and augmented.    We thank the reviewer for this suggestion. The Table 2 has been re-named. As the aim of this Table is to summarize the documented effects of smoking status on immune system in a sex-independent manner and since no definitive data are available about specific influence of smoking habits on immune system according by gender, we have added the following sentence within the text:

    However, even if scientific studies describe  various cellular mechanisms responsible for immune-regulation, the distinct impact of cigarette smoke and different reaction of immune system according by gender  is largely unknown, as well as the exact molecular pathways underlying smoking-associated immune-pathology.

  3. Some publications containing information and references related to this article are suggested for the authors' review.

    We have included  two suggested  references